# The Fatal Clinical Outcome of Severe COVID-19 in Hospitalized Patients: Findings from a Prospective Cohort Study in Dhaka, Bangladesh

**DOI:** 10.3390/medicina59071280

**Published:** 2023-07-10

**Authors:** Tasnuva Ahmed, Afroza Akter, Imam Tauheed, Marjahan Akhtar, Sadia Isfat Ara Rahman, Fatema Khaton, Faisal Ahmmed, Md. Golam Firoj, Jannatul Ferdous, Mokibul Hassan Afrad, Zannat Kawser, Mohabbat Hossain, Mohammad Abul Hasnat, Mostafa Aziz Sumon, Asif Rashed, Shuvro Ghosh, Sayera Banu, Tahmina Shirin, Taufiqur Rahman Bhuiyan, Fahima Chowdhury, Firdausi Qadri

**Affiliations:** 1Infectious Diseases Division, International Centre for Diarrhoeal Disease Research Bangladesh (ICDDR,B), 68 Shaheed Tajuddin Ahmed Sarani, Mohakhali, Dhaka 1212, Bangladesh; tasnuva.ahmed@icddrb.org (T.A.); afrozaakter@icddrb.org (A.A.); imam.tauheed@icddrb.org (I.T.); marjahan.akhtar@icddrb.org (M.A.); sadia.rahman@icddrb.org (S.I.A.R.); fatema@icddrb.org (F.K.); faisal.ahmmed@icddrb.org (F.A.); golam.firoj@icddrb.org (M.G.F.); jannatulferdous271@gmail.com (J.F.); hassan.afrad@icddrb.org (M.H.A.); sbanu@icddrb.org (S.B.); taufiqur@icddrb.org (T.R.B.); fchowdhury@icddrb.org (F.C.); 2Institute for Developing Science & Health Initiatives (ideSHi), Dhaka 1216, Bangladesh; zannat@ideshi.org (Z.K.); mhossain.geb.cu@gmail.com (M.H.); 3Kurmitola General Hospital, Dhaka 1206, Bangladesh; drhasnat@gmail.com (M.A.H.); mostafa_sumon@yahoo.com (M.A.S.); 4Mugda Medical College & Hospital, Dhaka 1214, Bangladesh; asifrashed@gmail.com (A.R.); drshuvrosoumo413141@gmail.com (S.G.); 5Institute of Epidemiology, Disease Control and Research, Dhaka 1212, Bangladesh; tahmina.shirin14@gmail.com

**Keywords:** COVID-19, severe, deceased, antiviral, anticoagulant, Bangladesh

## Abstract

*Background and Objectives*: The morbidity and mortality associated with COVID-19 have burdened worldwide healthcare systems beyond their capacities, forcing them to promptly investigate the virus characteristics and its associated outcomes. This clinical analysis aimed to explore the key factors related to the fatal outcome of severe COVID-19 cases. *Materials and Methods*: Thirty-five adult severe COVID-19 patients were enrolled from two COVID-19 hospitals in Dhaka, Bangladesh. Clinical manifestation, comorbid conditions, medications, SARS-CoV-2 RT-PCR related cycle threshold (CT) value, hematology, biochemical parameters with SARS-CoV-2 specific IgG and IgM responses at enrollment were compared between the survivors and deceased participants. *Results*: Total 27 patients survived and 8 patients died within 3 months of disease onset. Deceased patients suffered longer from shortness of breath than the survived (*p* = 0.049). Among the severe cases, 62% of the deceased patients had multiple comorbid condition compared to 48% of those who survived. Interestingly, the anti-viral was initiated earlier among the deceased patients [median day of 1 (IQR: 0, 1.5) versus 6.5 (IQR: 6.25, 6.75)]. Most of the survivors (55%) received a combination of anticoagulant (*p* = 0.034). Liver enzymes, creatinine kinase, and procalcitonin were higher among the deceased patients during enrollment. The median CT value among the deceased was significantly lower than the survivors (*p* = 0.025). A significant difference for initial IgG (*p* = 0.013) and IgM (*p* = 0.030) responses was found between the survivor and the deceased groups. *Conclusions:* The factors including older age, male gender, early onset of respiratory distress, multiple comorbidities, low CT value, and poor antibody response may contribute to the fatal outcome in severe COVID-19 patients. Early initiation of anti-viral and a combination of anticoagulant treatment may prevent or lower the fatality among severe COVID-19 cases.

## 1. Introduction

The COVID-19 global pandemic has resulted in an important and critical threat to global health, since the outbreak in December 2019 in China. The spread of the infection has been very rapid. This has also extended rapidly in Bangladesh from March 2020 [1]. More than 1.5 million cases have been confirmed and 28,000 people have died from COVID-19 infection in Bangladesh [2]. In Bangladesh, the infectivity rate has ranged from 32% to 1.5% most recently and the case fatality rate (CFR) 1.56% has generally been lower than other countries [3].

The disease spectrum of COVID-19 in Bangladesh shows that about 20% suffer from moderate to severe disease with about 5–9% of the hospitalized patients requiring intensive care unit (ICU) admission [4]. High proportions of SARS-CoV-2 positive individuals, who are present as asymptomatic in the community, are probably responsible for the rapid spread of the infection [5]. The severity of COVID-19 is associated with the alteration of immune responses [6]. Lack of immunity to the virus elicits the pathogenesis of the disease process. Information on the mechanism of the immune response along with the clinical parameters is important to plan more effective diagnostic, therapeutic, and prophylactic strategies for combating SARS-CoV-2.

As the clinical spectrum of COVID-19 ranges from asymptomatic to severe diseases, severe disease may present as acute respiratory distress syndrome (ARDS) with high mortality risk [7,8], and there is a need for understanding factors that can be used as early markers of disease severity over the disease course. Current evidence suggests that patients with advanced age, respiratory distress, oxygen saturation less than 90%, or pre-existing comorbidities are more susceptible to becoming severe [9,10,11]. Different studies have shown that cardiovascular disease, diabetes, chronic obstructive pulmonary disease (COPD), and cerebrovascular disease are significant risk factors for COVID-19 severe cases [12]. The aim of this clinical analysis from a longitudinal cohort up to 3 months from study enrolment was to explore some of the key clinical features, comorbid conditions, and laboratory parameters of severe hospitalized COVID-19 cases during acute infection and their relation to the outcome either as a survivor or deceased.

## 2. Materials and Methods

### 2.1. Study Site and Participants

Hospitalized patients diagnosed as COVID-19 by reverse transcription-polymerase chain reaction (RT-PCR) carried out with nasopharyngeal swab (NPS) and classified as severe cases according to WHO criteria (respiratory distress with SpO_2_ < 90%) were included in this analysis from a longitudinal study conducted in Dhaka, Bangladesh [13]. A total of 35 adult severe patients were enrolled from two COVID-19 dedicated hospitals (Mugda Medical College & Hospital and Kurmitola General Hospital) from November 2020 to September 2021.

Follow-up of all the patients was continued either in these two hospitals or at a field site office in Mirpur after discharge, located in the northeast part of Dhaka city, where icddr,b has conducted many clinical trials and epidemiological studies.

The study protocol was approved by the institutional review board (IRB) of the International Centre for Diarrhoeal Disease Research (icddr,b) and the Directorate General of Health Services (DGHS). Written informed consent was obtained from all severe patients according to the ‘Declaration of Helsinki’ regulation and guideline.

### 2.2. Data and Sample Collection

Socio-demographic data, clinical symptoms, date of symptom onset, vital signs, comorbidities (obesity, diabetes mellitus, hypertension, heart disease, kidney disease, asthma, COPD, liver disease, immunodeficiency, malignancy, neurological disorder, and/or others) and medication received for the treatment of COVID-19 since symptom onset were collected during enrollment. Patients were enrolled on study day 1 and prospectively followed up on study days 7, 14, 28, and 90. We collected data on relevant clinical information (symptoms, vital signs, medication, hospitalization, and health outcome) during each follow-up visit. The date of the first symptom was considered as the date of COVID-19 disease onset.

Blood and NPS specimens were collected from all patients at all study day points. Clinical laboratory consisting of complete blood count (CBC), alanine aminotransferase, aspartate aminotransferase, creatinine, C-reactive protein (CRP), random blood sugar (RBS), lactate dehydrogenase (LDH), creatine kinase (CK), ferritin, and D-dimer and ABO blood group were analyzed from the specimen collected on study day 1, as these tests are usually performed for COVID-19 patients according to the national guideline of management of COVID-19 patients in Bangladesh [14]. Neutrophil lymphocyte ratio (NLR) was measured by the ratio of absolute neutrophil count to absolute lymphocyte count to explore the relatedness to alive and deceased [15,16]. SARS-CoV-2 specific antibody (IgM and IgG) concentration was also observed from blood samples collected during enrollment. Analyses were carried out at the icddr,b and the Institute for Developing Science and Health Initiatives (ideSHi) laboratories. All specimen were collected by trained study personnel as previously describes [11].

### 2.3. SARS-CoV-2 RT-PCR and Viral Load from NPS

Viral RNA was extracted using the magnetic bead-based Nexor 32 Fully Automated Nucleic Acid Extractor (Nucleic Acid Extraction or Purification Kit, Beijing Lepu Medical Technology Co., Ltd., Beijing, China). RT-PCR and estimated viral load were quantified using the Chinese CDC 2019-nCoV_N2 primers and probe set [17]. For the RT-PCR test, a positive test was set at a cycle threshold (CT value) of less than 40. The Human RNase P gene was used as an internal control to monitor viral RNA extraction efficiency [17]. Viral load was measured from the viral copy numbers quantified using N2 quantitative PCR standards [11]. The CT value and viral load from samples collected on day 1 was carried out to explore the impact of the viral load on the negative outcome in severe patients.

### 2.4. Immune Response Assessment

The IgM and IgG antibody responses specific to RBD in sera collected from severe patients on study day 1 had been done only in this analysis [18,19]. The RBD-specific antibody concentrations (ng/mL) were measured using an in-house ELISA (Enzyme Linked Immunosorbent Assay) assay for IgG and IgM isotypes against the receptor binding domain (RBD) of the spike protein of SARS-CoV-2 [19,20]. To determine the antibody concentration in serum, CR3022 was used to generate a standard curve for both IgG and IgM (starting concentration 25 ng/mL) which was validated using serum specimens collected from RT-PCR positive COVID-19 patients, pre-pandemic healthy controls (frozen sera specimens collected between 2015–2018), influenza infected patients and serum collected from surveillance of Japanese encephalitis and compared the responses with commercially available ELISA kits (Euroimmun, Germany; Wantai, China) [20]. To determine the cut-off for seropositivity, the median plus the range of concentrations of SARS-CoV-2 IgG and IgM antibodies measured in pre-pandemic serum samples were used. Based on this, the cut-off for seropositivity was set to 500 ng/mL (0.5 µg/mL) as the positive cut-off value for both IgG and IgM antibodies. We have described step by step procedure for carrying out this ELISA in earlier article [11,20].

### 2.5. Statistical Analysis

Demographic and clinical characteristics of severe patients were stratified by survival status (survivor and deceased) and measured as mean with 95% confidence interval (CI) for continuous data, median with inter-quartile range (IQR) for ordinal data, and percentage for categorical data. To identify the significant difference, *t*-test was used to compare the mean difference between the two groups while for median and percentage, non-parametric Mann Whitney U test and proportion test (z test) are used respectively. For the comparison of clinical laboratory results, we calculated median with IQR by survival status and Mann Whitney U test was used to measure the significant difference. Box plots with scattered points were created to see the distribution of clinical laboratory values in two groups. Geometric mean (GM) was calculated and *t*-test was used to compare antibody responses (IgG and IgM) between survivors and deceased severe cases at 95% level of significance. All analyses were done using R statistical software version 4.2.2 (“ggplot2” and “ggpubr” packages for the scatter and boxplot diagram and “dplyr”, “reshape2” packages for data).

## 3. Results

### 3.1. Sociodemographic and Clinical Characteristics

A total of 35 severe COVID-19 patients who met the inclusion criteria were included in this analysis of whom 27 patients survived and 8 patients died within 3 months from study enrolment (Figure 1). The outcome events of the deceased individuals including symptom onset, hospitalization and enrollment are illustrated in Figure 2. Out of the eight deceased, 2 patients died in more than 6 weeks.

The median age among the survived and the deceased patients was 50 (IQR: 42.5, 62.5) and 61 (IQR: 54.5, 62.7) years respectively. Sociodemographic and clinical characteristics with comorbidities are shown in Table 1. Among the 27 survivors and 8 deceased, 19 (70%) and 5 (63%) cases were male respectively. There was no difference in the distribution of median duration from disease onset until diagnosis and hospitalization between the survival and deceased groups. However, the median duration from symptom onset till hospitalization was 3 days [IQR: 2, 6] and 8 days [IQR: 6, 7.5] in the deceased and survived group respectively and there was a statistically significant difference in distribution between the two groups (*p* = 0.046). In additional, there was no statistical significance in ABO blood group distribution between survivors and deceased.

The majority of severe cases in both survived and deceased groups had fever as the first symptom at disease onset (74.1% and 62.5%). All the severe patients suffered from shortness of breath up to day 28 follow-up followed by fever (91.4%) and cough (88.6%). Both survivors (52%) and deceased (63%) patients suffered from more than 3 clinical symptoms throughout the duration of e disease. Deceased patients suffered a longer duration from shortness of breath than those who survived, 23 and 14 median days respectively, which was statistically significant (*p* = 0.049) (Figure 3). Vital signs during enrollment were similar in both groups Table 1.

Among the deceased, almost all the patients required care at the ICU of the COVID-19 hospitals compared to only 11% (*n* = 3) of severe patients who survived which was statistically significant (*p* = 0.0003). Ventilation was not required for any cases who survived while 2 out of 8 patients who died received ventilation (*p* = 0.007). The majority of the severe cases had comorbidities (83%), while 62% (*n* = 5) of the deceased patients had more than one comorbid condition compared to 48% (*n* = 13) of those who survived (Table 1). Diabetes was the most common comorbid disease among both groups (Figure 4) with a median duration of 8 years and 6 years in survivor and deceased groups respectively although there was no statistically significant difference in distribution (*p* = 0.954) (Table 1).

### 3.2. Medication Received

All severe patients who survived received antibiotics and anti-coagulants, while 86% received steroids and 81.5% received anti-histamine as a major part of COVID-19 treatment (Table 2). Among the deceased severe patients, all of them received antibiotics and anti-coagulants. Anti-viral was given to only 2 out of 8 deceased patients (25%) compared to 56% (*n* = 15) of the survivors. Only 13 (37%) and 10 (29%) of the severe cases took Ivermectin and Theophylline.

More than 50% of the patients who survived received multiple anti-coagulants simultaneously anytime during the disease course (Enoxaparin sodium and Rivaroxaban) compared to only 12.5% (*n* = 1) patients who died and the difference of this proportion is statistically significant (*p* = 0.032). The median duration between hospitalization and initiation of anti-viral treatment (Injection Remdisivir; 100 mL) was 1 day (IQR: 0, 1.5) and 6.5 days (IQR: 6.25, 6.75) among the survivors and deceased respectively although there was no statistical difference in distribution (*p* = 0.054). A single regimen of glucocorticoid steroid (one of either dexamethasone, methylprednisolone, hydrocortisone, or prednisolone) was given to 78.3% (*n* = 18) and 100% (*n* = 8) of the survivors and deceased patients respectively while patients who survived received multiple regimens of glucocorticoid steroid (*n* = 5). Although most of the severe cases received medication after hospitalization, 3 patients from the survival group received either antibiotic, anti-coagulant, and/or glucocorticoid steroid while none of the deceased patients received medication before hospitalization (Table 2).

### 3.3. Clinical Laboratory Findings

Clinical hematology and biochemistry findings of severe cases during enrollment are summarized in Table 3. The median value for differential neutrophil counts, RBS, LDH, CRP, and ferritin is higher than the normal range in both survival and deceased groups. Interestingly, liver enzymes such as alanine transaminase (ALT) and aspartate transaminase (AST) were both raised among the deceased group compared to the survival group with a median value of 68.5 (IQR: 27.5, 127.4) and 64 (IQR: 41.2, 101.5) versus 43 (IQR:25.3, 86.5) and 42 (IQR: 29, 57.8), respectively. Inflammatory biomarkers like CRP, CK, procalcitonin, and NLR during enrollment among the deceased patients were observed to be higher than those who survived, although the difference was not statistically significant (Figure 5).

### 3.4. Viral Load from NPS

Interestingly, the median CT value for the NPS RT-PCR value among the deceased patients at onset was found to be 21.11 (IQR: 19.1, 28.78) which was much lower than the value of 33.01 (IQR: 28.29, 34.71) seen in the patient who survived (*p* = 0.025) (Figure 6). The geometric mean of the viral copies analyzed from 24 severe cases was found higher among the deceased (*n* = 5) [94.64 × 10^4^ (95% CI: 27.58 × 10^2^–32.47 × 10^7^)] compared to the survivors (*n* = 19) [59.77 × 10^3^ (95% CI: 15.40 × 10^3^–23.19 × 10^4^)] although the difference was not significant (*p* = 0.191) Table 4.

### 3.5. SARS-CoV-2 Specific IgG and IgM Antibody Responses

We further evaluated the SARS-CoV-2 RBD specific antibody responses in all patients at the time of enrollment (Figure 7). Most patients who survived had antibody levels (93% for IgG and 74% for IgM). In contrast, among the deceased group, 25% patients were positive for IgG and 38% were positive for IgM. There was a significant difference in the proportion of those who were seropositive for IgG between survivors and deceased (*p* < 0.001) Table 4. We found a significant difference in geometric mean concentration for IgG antibody responses between the survivor and the deceased group (1686 ng/mL vs. 182 ng/mL, *p* = 0.013). Similarly, geometric mean concentration of IgM was also significantly higher (1306 ng/mL vs. 406 ng/mL, *p* = 0.030) in the survived patients compared to the patients who had deceased Table 4.

## 4. Discussion

The SARS-CoV-2 pandemic is the most destructive global event of this century. Scientists and physicians are struggling for different remedies and treatments including vaccine development to mitigate this infection. This study was focused on understanding the clinical and immunological differences between survivors and deceased severe COVID-19 cases. In Dhaka, the capital of Bangladesh, the Kurmitola General Hospital and Mugda Medical College and Hospital are the two major hospitals that are responsible for the treatment of critical COVID-19 patients from the very beginning of the pandemic since March 2020. From this cohort 23% (*n* = 8) of the severe patients died and most of the deceased patients died within one month from disease onset. This finding is very similar to the study conducted in Germany which revealed that 30% of severe COVID-19 patients died within 30 days [21]. Around 29% severe cases in our study were admitted in the ICU, among them 6% needed ventilation of whom 25% died. This data is also very similar to the study conducted in New York where 21% of severe patients died in the ICU [22]. The requirement of ICU and ventilation was related to the deceased group of our study participants which is also seen in previous studies [23,24,25]. The death rate among the older and male patients was higher in our study and comparable with the previous studies [26]. Compared to the survivors among severe cases, the duration from symptom onset to hospitalization was significantly shorter in the deceased group and this finding is also similar with a study conducted in Wuhan [23]. A plausible explanation from our analysis could be due to the early onset of shortness of breath and low CT value (assuming higher viral load) among the severe patients lead to fatal outcome. Fever was the most common initial symptom among 71% of severe COVID-19 patients and it is also the common symptom in patients with SARS or MERS [27]. None of the reported symptoms were related to the increased risk of death in our study.

There was no difference among the vital signs of the deceased and survivor group in this study but most of the deceased patients had multiple comorbidities. Evidence from meta-analysis revealed that diabetes and heart disease were the most common risk factors for adverse outcome of SARS-CoV-2 infection and the duration of the comorbidities also has a contribution to having a bad prognosis [12].

The physicians at the study hospitals practiced COVID-19 management according to the national guideline of management of COVID-19 patients in Bangladesh [14]. The majority of the severe patients in our study received combination of drugs: antibiotics, anticoagulants, and glucocorticoids or corticosteroids. Moreover, we found significant difference in the survivor group who received multiple anti-coagulants (Enoxaparin sodium and Rivaroxaban) compared to the deceased group. Earlier report suggests effective and timely administration of anticoagulant is effective for preventing fatal outcome of severe COVID-19 cases [28].

No difference was found in the hematological parameters (CBC, D-dimer, ferritin, and CRP) as all of the participants included in this analysis were severe cases. However, we observed a trend of higher than normal range for liver enzymes (ALT and AST), NLR, CK and procalcitonin level among the deceased group compared to survivors during acute infection. We found that the low CT value was significantly associated with the deceased patient and also the viral load was high in the deceased cases which suggests that the initial viral load was higher among the deceased than those who survived and hence required early hospitalization. From our previous study, we also observed that CT value was lower during the early stage of the disease onset [11]. We know that CT value has a strong association with the viral load and that a lower CT value indicates a higher viral load [29]. A Swedish study has also presented a significant association between viral load and mortality among hospitalized COVID-19 symptomatic cases [30].

The SARS-CoV-2 specific IgG and IgM from our cohort were detected as early as 4 days from disease onset. The results of this study suggest that higher levels of RBD specific IgG and IgM antibodies during acute infection were associated with patient recovery, since, the majority of the deceased patients did not respond to RBD-specific IgG and IgM antibody at the acute phase of the disease while most of the survivors seroconverted for both isotypes. In earlier studies, it was shown that IgG levels were higher in critical COVID-19 patients who survived suggesting that the antibody response correlates with virus neutralization, and functional protection [31]. Another study has shown that IgG responses were detected in most severe and mild COVID-19 patients within 9 days from disease onset, and remained high up to 4 weeks in severe patients [32]. COVID-19 patients generally show a rapid increase in SARS-CoV-2-specific IgM, and IgG, commonly observed around a week after the infection [11], however, comorbidities may also have an impact not only on the inflammatory response during COVID-19 but also on antibody production, as seen in earlier reports in human immunodeficiency virus (HIV) patients presenting a delayed SARS-CoV-2 specific IgM and IgG production [33].

One of the limitations in our study is that we required signed informed consent to enroll study participants and so the severe cases in this study were enrolled from the COVID-19 ward instead of ICU. Thus, no critical case (unconscious or on life support) were enrolled in the study which could have given us a better picture of the fatal outcome of COVID-19. A small sample size was another limitation for which there was not much power to carry out multivariable survival analysis like Cox regression to state the causal association between risk factors and fatal outcome.

## 5. Conclusions

In summary, we show differences in severe SARS-CoV-2 infected patients between those who succumbed to the disease and those who survived the infection. We feel a combination of factors including early onset and longer duration of respiratory distress, multiple comorbid conditions, low RT-PCR SARS-CoV-2 CT values, poor antibody responses, all may contribute to the death of observed severe cases. Early initiation of anti-viral medication with a combination of anticoagulant treatment may also prevent the fatal outcome for patients suffering from severe COVID-19. However, a larger cohort needs to be studied to further substantiate these findings.

## Figures and Tables

**Figure 1 medicina-59-01280-f001:**
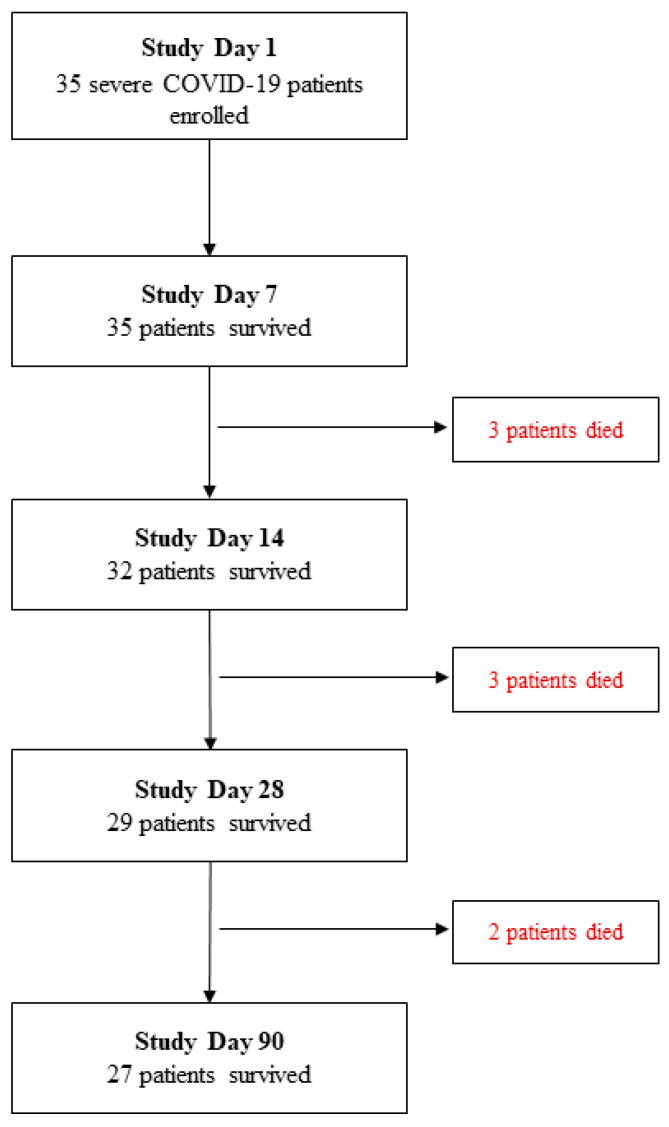
Severe COVID-19 patient enrollment and study follow-up flow chart.

**Figure 2 medicina-59-01280-f002:**
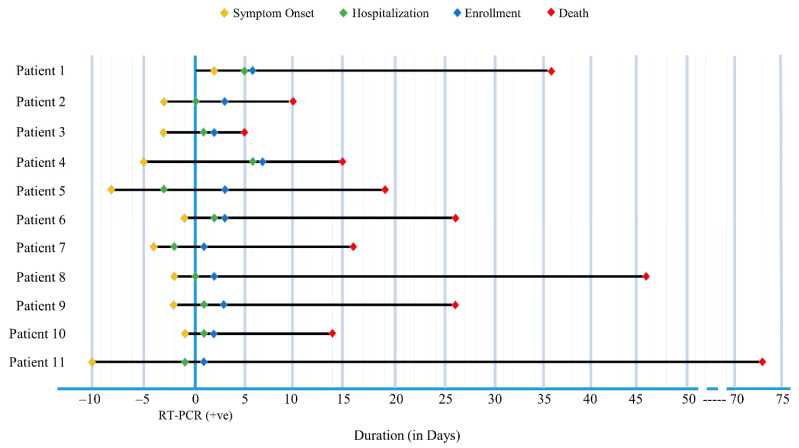
**Events of Deceased COVID-19 severe cases.** Eight deceased patients individually represented illustrating the events starting from symptom onset till death over the time (*X*-axis). The *Y*-axis indicated the date of the first SARS-CoV-2 RT-PCR positive nasopharyngeal swab (NPS) collected.

**Figure 3 medicina-59-01280-f003:**
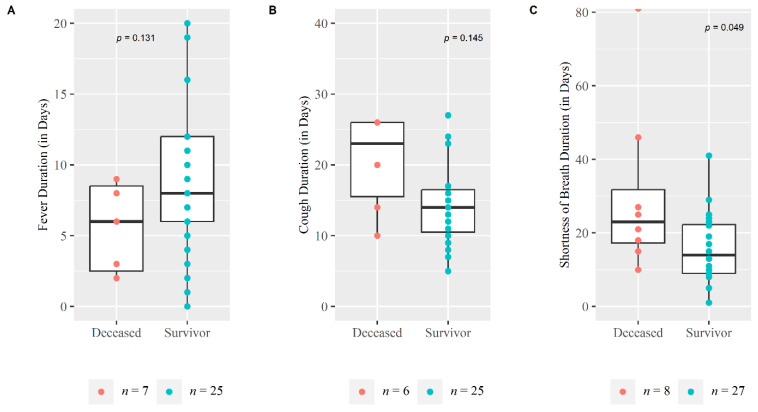
**Duration of symptoms among survival and Deceased Severe COVID-19 patients.** The Whisker boxplot and scatter points were plotted for Symptom duration in days (on *Y*-axis) against deceased and survivor groups (on *X*-axis): (**A**) Fever duration, (**B**) Cough duration and (**C**) Shortness of breath duration. Non-parametric Mann Whitney U test was performed to measure the difference in the symptom duration distribution of these two groups.

**Figure 4 medicina-59-01280-f004:**
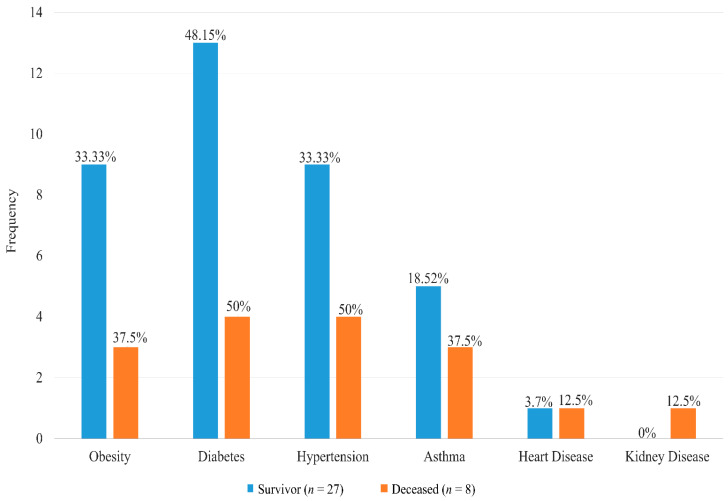
**Distribution of comorbid disease among the severe COVID-19 patients.** The bar diagram illustrating the frequency distribution with percentage of comorbidities among the deceased and survivor groups.

**Figure 5 medicina-59-01280-f005:**
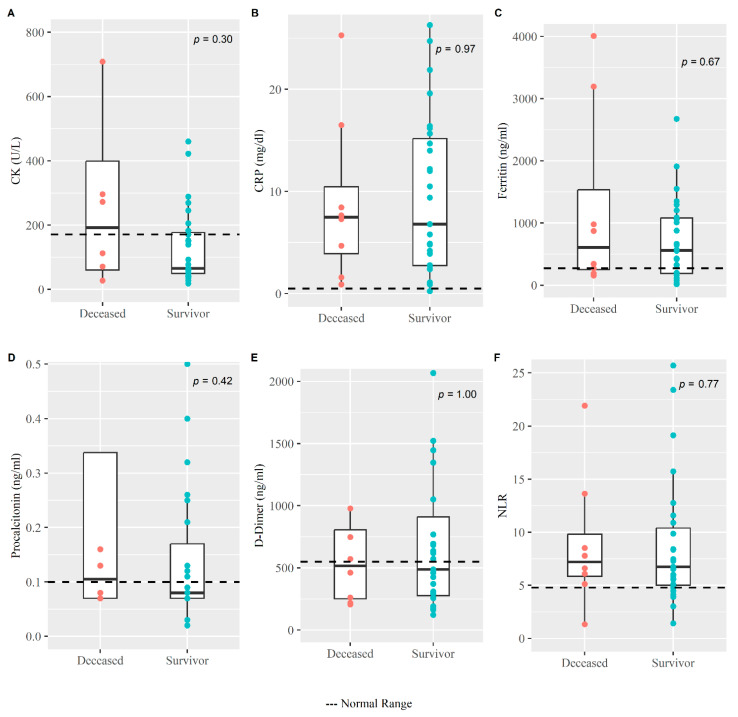
**Clinical inflammatory biomarkers among the survivors and deceased COVID-19 patients.** The Whisker boxplot for various clinical laboratory results with scatter points (on *Y*-axis) were plotted for deceased and survivor groups (on *X*-axis). The normal range for (**A**) CK, (**B**) CRP, (**C**) Ferritin, (**D**) Procalcitonin, (**E**) D-Dimer and (**F**) NLR was <171 U/L, <0.5 mg/dL, ≤274.66 ng/mL, <0.1 ng/mL, <550 ng/mL and <4.795 respectively. Non-parametric Mann Whitney U test was performed to measure the difference in the distribution of the laboratory results between these two groups.

**Figure 6 medicina-59-01280-f006:**
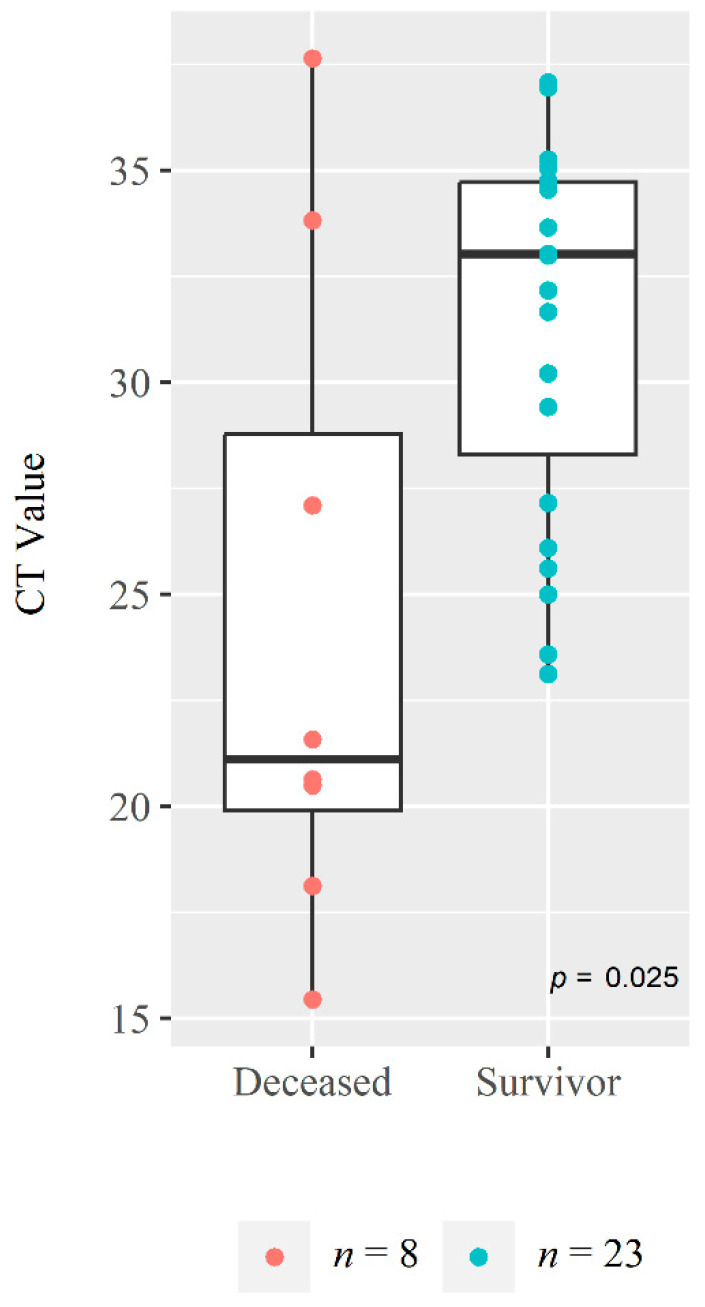
**Comparison of cycle threshold (CT) value at enrollment between deceased and survivors.** The Whisker boxplot of CT values with scatter points (on *Y*-axis) were shown for both deceased and survivor groups (on *X*-axis) and the difference in the distribution of the CT values between these two groups were measured using non-parametric Mann Whitney U test.

**Figure 7 medicina-59-01280-f007:**
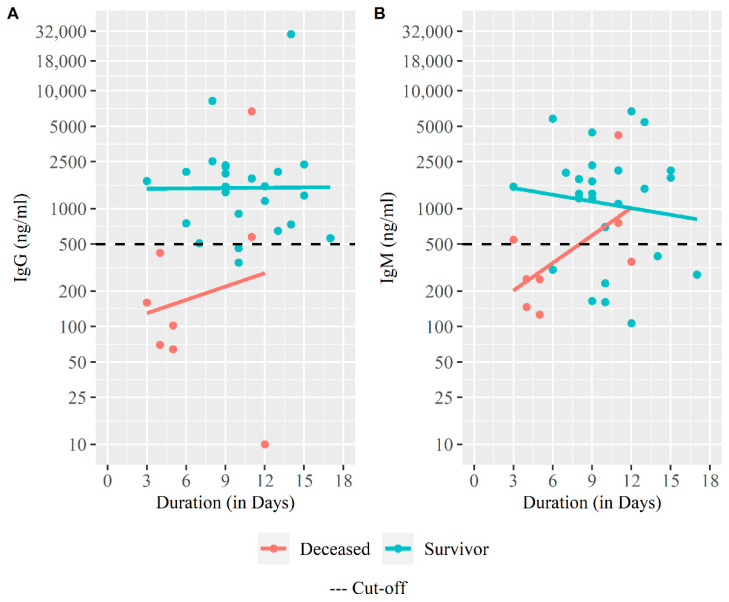
**SARS-CoV-2 specific IgG and IgM antibody response among severe COVID-19 patients.** (**A**) Trend of initial IgG antibody response among the deceased and survivors, (**B**) Trend of initial IgG antibody response among the deceased and survivors. The scatter plot with fitted linear regression line was drawn for antibody titer (on *Y*-axis) against duration between symptom onset and enrollment (on *X*-axis). The plot was created with log transformed *Y*-axis scale.

**Table 1 medicina-59-01280-t001:** Sociodemographic and Clinical characteristics of Severe COVID-19 patients.

Variables	All Patients *n* = 35	Survivor (*n* = 27)	Death (*n* = 8)	*p*-Values *
Age, Median (IQR)	55 (44, 62.5)	50 (42.5, 62.5)	61 (54.5, 62.75)	0.215
Sex, Male *n* (%)	24 (68.57%)	19 (70.37%)	5 (62.5%)	0.674
**Blood Group**				
O Blood Group *n* (%)	12 (34.29%)	9 (33.33%)	3 (37.5%)	0.827
Non-O Blood Group *n* (%)	23 (65.71%)	18 (66.67%)	5 (62.5%)	
**Duration interval in days, median (IQR) ^a^**				
Disease Onset and Diagnosis	5 (3, 7)	6 (3, 7)	3 (1.75, 5.75)	0.206
Disease Onset and Hospitalization	7 (5, 9)	8 (6, 7.5)	3 (2, 6)	0.046
Duration of hospitalization	14 (10, 22)	13 (10, 19.5)	19 (12.25, 28)	0.245
Disease Onset and Onset of Shortness of Breath	4 (1.5, 6.5)	4 (1.5, 7)	2 (1.5, 3.75)	0.303
**First symptom at disease onset, *n* (%) ^b^**				
Fever	25 (71.43%)	20 (74.07%)	5 (62.5%)	0.525
Cough	7 (20%)	5 (18.52%)	2 (25%)	0.687
Others (Loss of Taste, Runny Nose, Shortness of Breath)	3 (8.57%)	2 (7.41%)	1 (12.5%)	0.651
**Reported symptoms over the first 1 month, *n* (%) ^b^**				
Fever	32 (91.43%)	25 (92.59%)	7 (87.5%)	0.651
Cough	31 (88.57%)	25 (92.59%)	6 (75%)	0.169
Shortness of breath	35 (100%)	27 (100%)	8 (100%)	-
Others **	21 (60%)	16 (59.26%)	5 (62.5%)	0.869
**No. of symptoms reported over the first 1 month, *n* (%) ^b^**				
≤3 symptoms	16 (45.71%)	13 (48.15%)	3 (37.5%)	0.595
>3 symptoms	19 (54.29%)	14 (51.85%)	5 (62.5%)	
**Vital Signs, Mean (95% CI) ^c^**				
Systolic pressure (mmHg)	125.51 (119.1, 131.9)	128 (120, 136)	117 (106, 128)	0.079
Diastolic pressure (mmHg)	74.8 (70.86, 78.74)	76.1 (71.4, 80.7)	70.5 (62.3, 78.7)	0.200
Pulse rate (beats/min)	84.49 (80.4, 88.6)	85.5 (80.7, 90.4)	81 (71.6, 90.4)	0.347
Respiratory rate (times/min)	30.83 (28.5, 33.2)	31.2 (28.7, 33.7)	29.5 (22.5, 36.5)	0.604
Temperature (°F)	97.60 (97.4, 97.8)	97.6 (97.3, 97.8)	97.7 (97.4, 98)	0.477
SpO2% (during hospital admission)	83.51 (80.12, 86.91)	83.93 (80.81, 87.04)	82.13 (69.13, 95.12)	0.760
Oxygen flow at enrolment (L/min)	8.21 (6.81, 9.6)	8 (6.4, 9.6)	8.88 (5.32, 12.43)	0.616
**ICU required, n(%) ^b^**	10 (28.57%)	3 (11.11%)	7 (87.5%)	0.0003
**Ventilation required, n(%) ^b^**	2 (5.71%)	0	2 (25%)	0.007
**Comorbidities, *n*(%) ^b^**				
Yes	29 (82.86)	22 (81.48)	7 (87.5)	0.692
**No. of Comorbidities ^b^**				
≤1	17 (48.57%)	14 (51.85%)	3 (37.5%)	0.476
>1	18 (51.43%)	13 (48.15%)	5 (62.5%)	
**Duration (in years) of Comorbidities, Median (IQR) ^a^**				
Diabetes (*n* = 17, *n*_A_ = 13)	7 (4, 10)	8 (2,10)	6 (4.75, 9)	0.954
Heart disease (*n* = 2, *n*_A_ = 1)	10 (7.5, 12.5)	15 (15, 15)	5 (5, 5)	-
Hypertension (*n* = 13, *n*_A_ = 9)	7 (2, 10)	7 (2, 10)	7 (4.5, 9.25)	0.754
Asthma (*n* = 8, *n*_A_ = 5)	10 (7, 20.75)	10 (7, 20)	10 (8.5, 17.5)	0.763
Kidney disease (*n* = 1, *n*_A_ = 0)	0.0822 (0.08, 0.08)	-	0.0822 (0.08, 0.08)	-

Abbreviation: IQR—Interquartile range; SpO_2_—Oxygen Saturation; *n*_A_—No. of patients survived; CI: confidence interval * Comparison between survival and deceased group. ^a^ Mann Whitney U-test has been done for non-parametric distribution; *p* < 0.05 was considered statistically significant ^b^ Proportion Z-test done for data presented as proportion, No. (%); *p* < 0.05 was considered statistically significant ^c^ *t*-test done for comparing means between two groups; *p* < 0.05 was considered statistically significant ** Other symptoms include loss of taste, loss of smell, sore throat, runny nose, chest pain, muscle aches, joint pain, headache, vomiting, and diarrheal.

**Table 2 medicina-59-01280-t002:** Medication received for COVID-19 management.

Medication Received	All Patients *n* = 35	Survivor *n* = 27	Deceased *n* = 8	*p*-Values
**Antibiotic ^a^, *n* (%)**	35 (100%)	27 (100%)	8 (100%)	**-**
Single	6 (17.14%)	4 (14.81%)	2 (25%)	0.502
Multiple	29 (82.86%)	23 (85.19%)	6 (75%)	
Interval between Hospitalization and antibiotic start, Median(IQR)	0 (0, 4)	0 (0, 4)	1.5 (0, 5)	0.289
**Anti-Coagulant ^b^, *n* (%)**	35 (100%)	27 (100%)	8 (100%)	-
Single	19 (54.29%)	12 (44.44%)	7 (87.5%)	0.032
Multiple	16 (45.71%)	15 (55.56%)	1 (12.5%)	
Interval between Hospitalization and anti-Coagulant start, Median(IQR)	0 (0, 9)	0 (0, 10)	0 (0, 1)	0.402
**Anti-histamine ^c^, *n* (%)**	28 (80%)	22 (81.48%)	6 (75%)	0.687
Single	22 (78.57%)	17 (77.27%)	5 (83.33%)	0.748
Multiple	6 (21.43%)	5 (22.73%)	1 (16.67%)	
Interval between Hospitalization and anti-histamine start, Median(IQR)	0 (0, 1)	0 (0, 1)	0 (0, 1)	0.783
**Anti-helminthic (Ivermectin), *n* (%)**	13 (37.14)	10 (37.04)	3 (37.5)	0.981
Interval between Hospitalization and Ivermectin start, Median(IQR)	0 (0, 0)	0 (0, 0)	0 (0, 0)	1.00
**Anti-viral ^d^, *n* (%)**	17 (48.57)	15 (55.56) *****	2 (25)	0.129
Interval between Hospitalization and Anti-Viral start, Median(IQR)	1 (0, 2)	1 (0, 1.5)	6.5 (6.25, 6.75)	0.054
**Theophylline (Doxofylline), *n* (%)**	10 (28.57)	9 (33.33)	1 (12.5)	0.252
Interval between Hospitalization and theophylline start, Median(IQR)	6 (2.25, 9.75)	4 (2, 9)	10 (10, 10)	0.484
**Glucocorticoid steroid ^e^, *n* (%)**	31 (88.57%)	23 (85.19%)	8 (100%)	0.247
Single	26 (83.87%)	18 (78.26%)	8 (100%)	0.569
Multiple	5 (16.13%)	5 (21.74%)	0 (0%)	
Duration between Hospitalization and steroid start, Median(IQR)	1 (0, 9.25)	1 (0, 10.25)	0.5 (0, 1.50)	0.324
**Treatment Initiation before Hospitalization *n* (%) ***	3 (8.57%)	3 (11.11%)	0 (0%)	0.324

Abbreviation: IQR—Interquartile range; Proportion Z-test done for data presented as proportion, No. (%); Mann Whitney U-test has been done for non-parametric distribution; *p* < 0.05 was considered statistically significant ^a^ Antibiotics are given either oral or parenteral: Azithromycin, meropenem, doxycycline, ceftriaxone, moxifloxacin, amoxicillin+clavulanic acid, ciprofloxacin, cefixime, cefuroxime, cephradine, clarythromycin, linezolid. ^b^ Anti-coagulant drugs taken: Enoxaparin sodium and/or Rivaroxaban ^c^ Anti-histamine h1 receptor antagonist: Chlorpheniramine maleate, Fexofenadine hydrochloride, Cetirizine ^d^ Anti-viral: Remdesivir ^e^ Glucocorticoid steroid: Dexamethasone, methylprednisolone, hydrocortisone, or prednisolone * Treatment initiation with either Antibiotic, Anti-Coagulant or Glucocorticoid.

**Table 3 medicina-59-01280-t003:** Clinical hematology and biochemistry result of severe COVID-19 cases during enrollment.

Variables Normal Reference	All Patient *n* = 35	Survivor (*n* = 27)	Death (*n* = 8)	*p*-Values *
TLC (10^9^/L) 4.0–11.0	8.12 (6.06, 9.76)	7.79 (6.06, 9.76)	9.31 (7.34, 9.99)	0.569
Diff. Neu (%) 40–75	81.30 (77.40, 84.95)	81.30 (77.25, 84.95)	82.10 (80.90, 84.65)	0.492
Diff. Lymph (%) 20–45	11.80 (8.00, 15.20)	11.80 (8.00, 15.20)	11.50 (8.90, 13.93)	0.922
Diff. Mono (%) 2–10	5.50 (3.95, 8.65)	6.80 (4.30, 8.95)	4.50 (3.38, 5.90)	0.074
Diff. Eosino (%) 1–6	0.00 (0.00, 0.2)	0.00 (0.00, 0.4)	0.00 (0.00, 0.00)	0.072
Diff. Baso (%) 0.01–0.1	0.1 (0.1, 0.25)	0.1 (0.1, 0.35)	0.1 (0.1, 0.125)	0.368
Abs. Neut (10^9^/L) 2.0–7.5	6.74 (4.96, 8.68)	5.97 (4.96, 8.68)	7.59 (6.04, 8.595)	0.556
Abs. Lymph (10^9^/L) 1.5–4.0	0.83 (0.65, 1.195)	0.83 (0.62, 1.16)	0.9 (0.74, 1.24)	0.492
Abs. Mono (10^9^/L) 0.2–0.8	0.44 (0.36, 0.63)	0.44 (0.37, 0.64)	0.46 (0.22, 0.52)	0.346
Abs. Eos (10^9^/L) 0.04–0.45	0.00 (0.00, 0.015)	0.00 (0.00, 0.025)	0.00 (0.00, 0.00)	0.0589
Abs. Bas (10^9^/L) 0.01–0.1	0.01 (0.01, 0.02)	0.01 (0.01, 0.02)	0.01 (0.01, 0.01)	0.406
Plt (10^9^/L) 150–450	266 (208, 365.5)	282 (215.5, 371.5)	207 (155.2, 298)	0.107
RBS (mmol/L) 4.20–7.80	10.71 (7.29, 14.24)	10.71 (7.69, 13.28)	11.47 (7.32, 14.95)	0.738
Cr (µmol/L) 64–104	79.80 (64.82, 100.01)	78.01 (63.48, 97.57)	85.05 (74.19, 111.25)	0.492
ALT (U/L) <50	43 (25.34, 101.50)	43 (25.34, 86.50)	68.50 (27.50, 127.40)	0.709
AST (U/L) <50	47 (29, 61.50)	42 (29, 57.8)	64 (41.23, 101.50)	0.121
LDH (U/L) <248	401.2 (306.1, 485.5)	401.2 (318, 485.5)	373.1 (271.7, 491.5)	0.768
CK (U/L) <171	77.03 (50.05, 225.65)	65.42 (50.05, 176.87)	192.54 (60.33, 399.1)	0.298
CRP (mg/dL) <0.5	7.30 (2.75, 15.185)	6.80 (2.75, 15.19)	7.485 (3.91, 10.46)	0.969
Ferritin (ng/mL) ≤274.66	561.9 (190.8, 1082.8)	561.9 (190, 1082.8)	608.6 (251, 1534)	0.670
PCT (ng/mL) <0.1	0.08 (0.07, 0.185)	0.08 (0.07, 0.17)	0.105 (0.07, 0.338)	0.417
D-Dimer (ng/mL) <550	488 (260.5, 873.5)	488 (276.5, 910)	516.7 (251.2, 805.7)	1.000

Data displayed as median (IQR: interquartile range); * Mann Whitney U-test was done between survival and deceased group; *p*-value < 0.05 is considered as a significant difference in distribution. Abbreviation: Total leucocyte count (TLC), Differential neutrophil count (Diff. Neut), Differential lymphocyte count (Diff. Lymph), Differential monocyte count (Diff. Mono), Differential eosinophil count (Diff. Eos), Differential basophil count (Diff. Baso), Absolute neutrophil count (Abs.Neut), Absolute lymphocyte count (Abs.Lymph), Absolute monocyte count (Abs. Mono), Absolute eosinophil count (Abs. Eos), Absolute basophil count (Abs. Bas), Platelet (Plt), Random blood sugar (RBS), Alanine aminotransferase (ALT), Aspartate aminotransferase (AST), Creatinine (Cr), C-reactive protein (CRP), Lactate dehydrogenase (LDH), Creatine Kinase (CK) and Procalcitonin (PCT).

**Table 4 medicina-59-01280-t004:** Comparison of viral load and antibody response between Survivors and Deceased COVID-19 patients.

Variables	All Patients *n* = 35	Survivor (*n*_S_ = 27)	Deceased (*n*_D_ = 8)	*p*-Values ^a^
Viral Load (*n* = 24), GM (95% CI) *	10.63 × 10^4^ (25.62 × 10^3^, 44.08 × 10^4^)	59.77 × 10^3^ (15.40 × 10^3^, 23.19 × 10^4^)	94.64 × 10^4^ (27.58 × 10^2^, 32.47 × 10^7^)	0.191
**IgG (ng/mL)**				
≥500, *n* (%)	27 (77.14%)	25 (92.59%)	2 (25%)	<0.001
GM (95% CI)	1013.38 (585.98, 1752.51)	1685.59 (1101.55, 2579.29)	181.96 (36.55, 905.84)	0.013
**IgM (ng/mL)**				
≥500, *n* (%)	23 (65.71%)	20 (74.07%)	3 (37.5%)	0.056
GM (95% CI)	999.58 (607.79, 1643.93)	1305.58 (737.98, 2309.72)	405.85 (158.78, 1037.40)	0.030

* Viral load was measured for 19 survivors and 5 deceased patients ^a^ Proportion Z-test done for data presented as proportion, no. (%) and *t*-test done for comparing geometric means (GM) with 95% confidence interval (CI) between two groups; *p* < 0.05 was considered statistically significant.

## Data Availability

Data cannot be shared publicly because they are confidential. Data are available from the respective department of icddr,b for researchers who meet the criteria for access to confidential data.

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
