# Peer review of "The Fatal Clinical Outcome of Severe COVID-19 in Hospitalized Patients: Findings from a Prospective Cohort Study in Dhaka, Bangladesh"

_medicina, 2023, doi:10.3390/medicina59071280_

Round 1

Reviewer 1 Report

Dear authors,

This cohort study aimed to evaluate factors associated with mortality in severe COVID-19 patients. The study included a small number of participants, with 35 patients, of which 8 patients died. The overall methodology is acceptable; however, the study's major limitation is the small sample size and the statistical methods employed. Given the heterogeneity between survivor and deceased participants, it is crucial to adjust for confounding factors using multivariable analysis, such as Cox proportional hazards model or other parametric models, to account for important confounders. With a small sample size, overfitting of the model may occur, leading the authors to use univariable analysis. However, univariable analysis alone cannot establish a causal association between the factors of interest and mortality outcomes.

Another issue is that in Table 1, the PaO2/FiO2 ratio should be provided to compare the severity between the two groups.

The use of the log-rank test in Figure 4 is not appropriate due to differences in baseline characteristics. It would be more appropriate to employ a survival model with adjusted confounding factors.

Overall, to strengthen the study's findings, it is recommended to address the limitations by utilizing multivariable analysis, considering important confounders, and providing a more comprehensive analysis of the severity of illness and adjusted survival models.

Best regards,

Author Response

Responses to Reviewer's 1 comments

Dear authors,

This cohort study aimed to evaluate factors associated with mortality in severe COVID-19 patients. The study included a small number of participants, with 35 patients, of which 8 patients died. The overall methodology is acceptable; however, the study's major limitation is the small sample size and the statistical methods employed.

1. Given the heterogeneity between survivor and deceased participants, it is crucial to adjust for confounding factors using multivariable analysis, such as Cox proportional hazards model or other parametric models, to account for important confounders. With a small sample size, overfitting of the model may occur, leading the authors to use univariable analysis. However, univariable analysis alone cannot establish a causal association between the factors of interest and mortality outcomes.

Response: Due to the small sample size, we did not get any significant association between sociodemographic and clinical characteristics with the mortality outcomes in bivariate analysis (Table 1 in the manuscript). However, we tried to carry out univariable analysis but did not find any significant causal association. Hence, we did not proceed with multivariable modelling. We could only carry out exploratory analysis to look into factors related to fatal outcome of severe COVID-19 patients. So, we also agree with you that we cannot establish causal association between factors of interest and mortality outcomes due to small sample size which we have discussed in our study analysis limitation (line 417-419).

2. Another issue is that in Table 1, the PaO2/FiO2 ratio should be provided to compare the severity between the two groups.

 Response: We did not have data on PaO2/FiO2 in our dataset hence this information is not included in our analysis.

 3. The use of the log-rank test in Figure 4 is not appropriate due to differences in baseline characteristics. It would be more appropriate to employ a survival model with adjusted confounding factors.

Response: Since we didn’t get any significant association in bivariate analysis shown in table 1 and did not conduct any multivariable survival model due to the small sample size, we have just simply shown the Kaplan-Meier curve and log-rank test to identify the difference in survival probability according to the number of comorbidity.

4. Overall, to strengthen the study's findings, it is recommended to address the limitations by utilizing multivariable analysis, considering important confounders, and providing a more comprehensive analysis of the severity of illness and adjusted survival models.

Response: As we mentioned above, due to the small sample size, we did not get any significant association in our univariable analysis including the important confounders (age, sex and presence of comorbidities) and other clinical characteristics with the mortality outcomes (Table 1 in the manuscript). Hence, we did not carry out multivariable modelling. However, we have tried to enhance our analysis by exploring the differences in clinical and laboratory parameters among the survived and deceased severe COVID-19 patient groups.

Please see the attached revised manuscript with highlight changes

Reviewer 2 Report

This is a well-conducted cohort study where the authors demonstrate the possible correlation between severe cases of COVID-19 and mortality versus survival. Although the sample size is small, which was discussed as a limitation of the study, the authors managed to identify significant differences between the two study groups, thus raising highly relevant hypotheses.

 However, as this is a cohort study, it is suggested that the authors include a flow diagram and provide more emphasis on summarizing the follow-up time. Additionally, it is not clear to the reader which antibody quantification assay the authors used. I recommend specifying the analysis method, such as enzyme-linked immunosorbent assay (ELISA), and not just referencing the article.

 It is necessary to include the meanings of the abbreviations used in the captions of the tables and figures, even if they are mentioned in the main text of the article. Figures and tables should be self-explanatory.

 The y-axis of Figure 6 is inverted, and I suggest arranging it in ascending order for better understanding of the graph.

 These are some suggestions to enhance the article, ensuring clarity and accuracy of the presented information.

Author Response

Reviewer 2

This is a well-conducted cohort study where the authors demonstrate the possible correlation between severe cases of COVID-19 and mortality versus survival. Although the sample size is small, which was discussed as a limitation of the study, the authors managed to identify significant differences between the two study groups, thus raising highly relevant hypotheses.

1. However, as this is a cohort study, it is suggested that the authors include a flow diagram and provide more emphasis on summarizing the follow-up time. Additionally, it is not clear to the reader which antibody quantification assay the authors used. I recommend specifying the analysis method, such as enzyme-linked immunosorbent assay (ELISA), and not just referencing the article.

Response: We have now added a flow diagram (Figure 1) summarizing the no. of participants completing the follow-up time points.

2. It is necessary to include the meanings of the abbreviations used in the captions of the tables and figures, even if they are mentioned in the main text of the article. Figures and tables should be self-explanatory.

Response: We have checked the table and figure captions and edited as required.

3. The y-axis of Figure 6 is inverted, and I suggest arranging it in ascending order for better understanding of the graph.

Response: We have now arranged the y-axis of Figure 6 (now Figure 7) in ascending order as per your suggestion.

4.These are some suggestions to enhance the article, ensuring clarity and accuracy of the presented information.

Response: We have now made some editing in the revised manuscript for clarity.

Please see the attachment of revised manuscript with highlight changes

Round 2

Reviewer 1 Report

Dear Authors,

Thank you for your revision of the manuscript. I appreciate your efforts in addressing the comments and limitations. However, I would like to point out that the use of the log-rank test in Figure 4 may not be appropriate.

Best regards,

Author Response

Thank you for your revision of the manuscript. I appreciate your efforts in addressing the comments and limitations. However, I would like to point out that the use of the log-rank test in Figure 4 may not be appropriate.

Responses: We aimed to explore the survival rate among severe COVID-19 patients with multiple and single/no comorbidity and conducted log rank test to test the significant difference between the groups (Kleinbaum, D.G. and Klein, M. (2005). Survival Analysis. A self-learning text. Springer). However, as we have discussed in our limitation (line 406-408), the sample size does not have much power to do any adjusted survival modeling like Cox-regression or Kaplan-Meier and so we are removing Figure 5 (Survival plot of severe COVID-19 patients with comorbidities) in the revised manuscript.
